# Osteoprotegerin Is more than a Possible Serum Marker in Liver Fibrosis: A Study into Its Function in Human and Murine Liver

**DOI:** 10.3390/pharmaceutics12050471

**Published:** 2020-05-21

**Authors:** Adhyatmika Adhyatmika, Leonie Beljaars, Kurnia S. S. Putri, Habibie Habibie, Carian E. Boorsma, Catharina Reker-Smit, Theerut Luangmonkong, Burak Guney, Axel Haak, Keri A. Mangnus, Eduard Post, Klaas Poelstra, Kim Ravnskjaer, Peter Olinga, Barbro N. Melgert

**Affiliations:** 1Department of Pharmacokinetics, Toxicology, and Targeting, Groningen Research Institute for Pharmacy, University of Groningen, 9713 AV Groningen, The Netherlands; adhyatmika@ugm.ac.id (A.A.); ceboorsma@gmail.com (C.E.B.); c.reker-smit@rug.nl (C.R.-S.); b.guney@student.rug.nl (B.G.); a.haak@student.rug.nl (A.H.); k.a.mangnus@student.rug.nl (K.A.M.); e.post@rug.nl (E.P.); k.poelstra@rug.nl (K.P.); 2Department of Pharmaceutics, Faculty of Pharmacy, Gadjah Mada University, Yogyakarta 55281, Indonesia; 3Department of Pharmaceutical Technology and Biopharmacy, Groningen Research Institute for Pharmacy, University of Groningen, 9713 AV Groningen, The Netherlands; e.beljaars@rug.nl (L.B.); kurnia.putri@farmasi.ui.ac.id (K.S.S.P.); theerut.lua@mahidol.ac.th (T.L.); p.olinga@rug.nl (P.O.); 4Faculty of Pharmacy, University of Indonesia, Depok 16424, Indonesia; 5Department of Molecular Pharmacology, Groningen Research Institute for Pharmacy, University of Groningen, 9713 AV Groningen, The Netherlands; h.habibie@rug.nl; 6Faculty of Pharmacy, Hasanuddin University, Makassar 90245, Indonesia; 7Groningen Research Institute for Asthma and COPD (GRIAC), University Medical Center Groningen, 9713 GZ Groningen, The Netherlands; 8Faculty of Pharmacy, Mahidol University, Bangkok 73170, Thailand; 9Department of Biochemistry and Molecular Biology, University of Southern Denmark, DK-5230 M Odense M, Denmark; ravnskjaer@bmb.sdu.dk

**Keywords:** cirrhosis, TGFβ1, CCl_4_, resolution, hepatic stellate cells, osteoprotegerin, RANKL, TRAIL

## Abstract

Osteoprotegerin (OPG) serum levels are associated with liver fibrogenesis and have been proposed as a biomarker for diagnosis. However, the source and role of OPG in liver fibrosis are unknown, as is the question of whether OPG expression responds to treatment. Therefore, we aimed to elucidate the fibrotic regulation of OPG production and its possible function in human and mouse livers. OPG levels were significantly higher in lysates of human and mouse fibrotic livers compared to healthy livers. Hepatic OPG expression localized in cirrhotic collagenous bands in and around myofibroblasts. Single cell sequencing of murine liver cells showed hepatic stellate cells (HSC) to be the main producers of OPG in healthy livers. Using mouse precision-cut liver slices, we found OPG production induced by transforming growth factor β1 (TGFβ1) stimulation. Moreover, OPG itself stimulated expression of genes associated with fibrogenesis in liver slices through TGFβ1, suggesting profibrotic activity of OPG. Resolution of fibrosis in mice was associated with decreased production of OPG compared to ongoing fibrosis. OPG may stimulate fibrogenesis through TGFβ1 and is associated with the degree of fibrogenesis. It should therefore be investigated further as a possible drug target for liver fibrosis or biomarker for treatment success of novel antifibrotics.

## 1. Introduction

Various causes of chronic damage to the liver, such as viral infections, drug toxicity, biliary problems, and high alcohol and/or fat consumption can lead to fibrosis and even cirrhosis. This process is characterized by an abundant deposition of extracellular matrix in liver tissue hampering normal liver functions [1,2]. Transplantation is now the only solution when the disease has fully developed [3,4]. To date, reliable fibrosis biomarkers to diagnose disease stage in patients are scarce, especially in the early phase when pharmacological treatment is still a possible option. Another consequence of the lack of good biomarkers is the difficulty in measuring therapeutic success of antifibrotic drug candidates in patients and in vivo in preclinical studies [5,6].

Several studies have pointed towards osteoprotegerin (OPG) as a novel clinical biomarker associated with liver diseases [7]. Higher serum levels were measured in patients with alcoholic liver cirrhosis, primary biliary cirrhosis, and nonalcoholic steatohepatitis, and OPG was found to correlate with disease severity in some of these studies [8,9,10,11,12,13,14,15]. Furthermore, OPG was included as an additional parameter in a panel of markers in the Coopscore© to increase the diagnostic accuracy of this test [16]. 

OPG, also known as tumor necrosis factor receptor superfamily member 11B (gene name *TNFRSF11B*), is a decoy receptor for RANKL (receptor activator of nuclear factor kappa-Β ligand) and TRAIL (TNF-related apoptosis-inducing ligand) [17]. OPG is known as one of the key factors of osteogenesis and is produced by osteoblasts to control osteoclast activity [18]. However, recent studies indicate that its activities are not confined to bone homeostasis, but may be more diverse and include a role in several organ pathologies, especially fibrosis [19,20] and tumor development [21,22]. Based on its scavenging activities of RANKL and TRAIL, OPG may be able to modulate fibrosis development through these ligands. However, current knowledge on the role of OPG in liver fibrosis, especially on a cellular and tissue level, is limited.

High OPG serum levels found in patients with liver fibrosis can originate from multiple organs and thus do not give any information about liver- or cell-specific regulation of OPG. A recent study reporting cell-type-resolved proteomic analysis of human liver showed that OPG is exclusively produced by hepatic stellate cells in healthy liver tissue, but no proteomic data are available yet from fibrotic livers [23]. Liu and colleagues found that (myo)fibroblasts are the main source of OPG production in cardiac fibrosis [24]. However, epithelial and smooth muscle cells have also been shown to produce OPG [25,26]. Therefore, the source of OPG in liver fibrosis is still unclear.

There is also only limited data available about the role of OPG in fibrogenesis and fibrolysis in general and particularly in liver fibrosis. Most studies have investigated OPG in the context of bone metabolism and the osteoporosis that often accompanies patients with cirrhosis. However, the profibrotic cytokines like TGFβ1, IL4, and IL17 were found to be stimulators of OPG production by (synovial) fibroblasts, whereas the antifibrotic cytokine IFNγ could inhibit OPG production by these cells [27,28]. Furthermore, a study by Toffoli et al. showed that OPG could induce the expression of fibronectin, collagen type I, III, and IV, as well as TGFβ1 in vascular smooth muscle cells in vitro and that TGFβ1 induced the expression and triggered the release of endogenous OPG in these cells [20]. These data suggest that OPG is associated with fibrosis in general through TGFβ1 and may even stimulate development of organ fibrosis. In this study, we now aimed to study the association with liver fibrosis more specifically by investigating (1) liver-specific production and expression of OPG during fibrosis development in vivo and in vitro; (2) the fibrotic regulation of OPG production in liver tissue; (3) the response of OPG production to spontaneous resolution and drug (IFNγ)-induced resolution of fibrosis.

## 2. Materials and Methods 

### 2.1. Animals

Male Balb/c mice (20–22 grams, used for the CCl_4_ liver injury model) and male and female C57BL/6 mice (18–28 grams, used for precision-cut liver slices) were obtained from Harlan (Horst, The Netherlands) and were kept in cages with 12 h of a light/dark cycle and received food and water ad libitum. The Institutional Animal Care and Use Committee of the University of Groningen approved the use of animals in this study (DEC5429 for CCl_4_ model and DEC6416AA for precision-cut liver slices).

### 2.2. Human Liver Tissue

Anonymized residual human liver tissue samples, obtained from the Department of Hepato-Pancreato-Biliary Surgery and Liver Transplantation (University Medical Center Groningen (UMCG), the Netherlands), were collected for further analysis. In the UMCG, all patients eligible for organ transplantation are asked to sign a general consent form for the use of left-over body material (after diagnostic procedures) for research purposes. The experimental protocols were approved by the Medical Ethical Committee of the UMCG and the anonymized tissue samples were used according to Dutch guidelines (www.federa.org). Control human liver tissue (n = 5 for tissue lysates and n = 6 for precision-cut slices) was obtained from residual liver tissue from patients undergoing partial hepatectomy because of metastasis of colorectal carcinoma and from donor organs unsuitable for transplantation or resized donor organs. Cirrhotic human liver tissue (n = 8 for tissue lysates and n = 7 for precision-cut slices) was obtained from patients undergoing liver transplantation. Indications for transplantation were primary sclerosing cholangitis, primary biliary cirrhosis, congenital cirrhosis, and Wilson’s cirrhosis. No other patient characteristics were available because anonymous tissue was used.

### 2.3. Liver Fibrosis Model

Male Balb/c mice were treated with increasing doses of CCl_4_ (Fisher Scientific, Waltham, MA, USA) in olive oil intraperitoneally twice a week: first week 0.5 mL/kg, second week 0.8 mL/kg, third until eighth week 1 mL/kg. Mice were sacrificed at week 8 after developing measurable fibrosis (n = 12). Control mice were sham-treated with olive oil for 8 weeks and served as healthy controls (n = 11). 

For the spontaneous resolution model, mice were treated with CCl_4_ for 4 weeks and were then allowed to recover for 1 week and sacrificed at week 5 (n = 6) as described before [29]. Control mice for this experiment were treated with CCl_4_ for 4 weeks and then immediately sacrificed (n = 6). 

For the drug-induced resolution model, mice were treated with CCl_4_ for 8 weeks and during weeks 7 and 8 mice were additionally treated with 2.5 μg/mice of IFNγ (Peprotech, Rocky Hill, CT, US), three times a week as described before (n = 6) [28]. Control mice were co-treated with saline (n = 8) and both groups were sacrificed after 8 weeks of CCl_4_ treatment. In all experiments, serum and livers were collected for further analyses. 

### 2.4. Precision-Cut Liver Slices

Murine precision-cut liver slices were prepared according to standard protocols described before [30]. Slices were either not incubated (controls) or incubated in triplicate with 5 ng/mL TGFβ1 (Peprotech), 10 μM galunisertib (Selleckchem, Munich, Germany) in combination with TGFβ1, 10 ng/mL OPG (R&D Systems, Minneapolis, MN, USA), or 20 μg/mL of antibody antiRANKL, antiTRAIL, or both (antibodies-online.com) with culture medium replacements every 24 h for a total of 48 h (n = 5–8). All collected samples were immediately snap-frozen in liquid nitrogen before storage at −80 °C until further processing for analyses. Viability of the slices was assessed by measuring ATP content per milligram tissue using a bioluminescence assay kit (Sigma-Aldrich, St. Louis, MO, USA) as previously reported by Hadi et al. [31]. Slice supernatants were collected for OPG ELISA.

Human precision-cut liver slices were prepared in a similar way as described for murine slices. Before preparation, liver tissue was incubated in ice-cold University of Wisconsin organ preservation solution (UW-solution). Cores of human liver were made by using a 5-mm diameter biopsy-punch. Control and cirrhotic liver slices were incubated for 1 hr in the same medium as described for murine slices [30], then transferred to a new medium and incubated for a further 48 hrs, and this culture supernatant was subsequently used for measurement of excreted OPG.

### 2.5. Cell Culture

Primary human hepatic stellate cells (HHSteC, ScienCell, Carlsbad, CA, USA) were cultured in stellate cell medium containing 2% fetal bovine serum (FBS) and 1% of stellate cells growth supplement (ScienCell) in a 12-well plate initially coated with 10% human serum albumin (Sigma-Aldrich, St. Louis, MO, USA) to maintain quiescent state or uncoated to induce activation and transformation to myofibroblast-like cells and incubated with 5 ng/mL TGFβ1 (Peprotech) to simulate fibrosis. Culture supernatants were collected for OPG ELISA and cells were collected for a protein assay to correct for the number of cells in a well.

### 2.6. Generation of Liver Tissue Lysates

Human or mouse liver tissue was collected in a lysis and extraction buffer containing 25 mM Tris (Sigma-Aldrich), 10 mM sodium phosphate (Sigma-Aldrich), 150 mM NaCl (Sigma-Aldrich), 0.1% SDS (Sigma-Aldrich), 1% Triton-X 100 (Sigma-Aldrich), and protease inhibitor (Thermo Scientific, Waltham, MA, USA), snap frozen and stored at –80 °C until analysis. The tissue was then mixed with the buffer using mini-bead beater for 40 s and centrifuged for 1 h at 12,300× *g*. Supernatants were collected and used for ELISA.

### 2.7. OPG Analysis

OPG levels in tissue lysates, serum, and culture supernatants were measured using a murine or human OPG DuoSet® ELISA kit (Cat. No. DY459 and DY805 for mouse and human, respectively, R&D Systems) according to the instructions provided by the manufacturer.

### 2.8. mRNA Analysis

mRNA was isolated from three pooled slices per condition using a Maxwell® LEV Simply RNA Cells/Tissue kit (Promega, Madison, Wisconsin, US). Total mRNA concentration was quantified using a NanoDrop® ND-1000 Spectrophotometer (Thermo Scientific) for cDNA synthesis using a Moloney Murine Leukemia Virus Reverse Transcriptase kit (Promega) in a Mastercycler® Gradient (Eppendorf, Hamburg, Germany) with the program 10 min at 20 °C, 30 min at 42 °C, 12 min at 20 °C, 5 min at 99 °C, and 5 min at 20 °C. Quantitative real-time PCR (qPCR) analysis was performed in triplicate with the synthesized cDNA to measure transcription of β-actin, Collagen-1α1 (Col1α1), α-smooth muscle actin (αSMA), Fibronectin (Fn1), TGFβ1, heat shock protein-47 (HSP47) and OPG using SensiMixTM SYBR® Green (Bioline, London, UK) in a 7900HT Real-Time PCR sequence detection system (Applied Biosystems, Waltham, MA, US) with primer sequences as presented in Appendix A, Table A1. qPCR analysis consisted of 45 cycles of 10 min at 95 °C, 15 s at 95 °C, and 25 s at 60 °C (repeated for 40 times) followed by dissociation stage of 95 °C for 15 s, 60 °C for 15 s, and 95 °C for 15 s. Output data were analyzed using SDS 2.4 software (Applied Biosystems) and ΔCt values were normalized to housekeeping gene β-actin and relative gene expression was calculated as 2^−ΔCt^.

### 2.9. Single-Cell RNA Sequencing and Analyses

Generation of these data has been described before [32]. In short, female C57BL6/J mice were treated with vehicle or CCl_4_ for 2 or 4 weeks (n = 3). Sinusoid-associated CD105- and F4/80-positive cells were isolated by flow cytometry and processed for single-cell RNA-sequencing using the 10x Genomics platform. Minor cell populations and doublet cells were removed from the dataset leaving three major Louvain clusters corresponding to hepatic stellate cells, liver endothelial cells, and Kupffer cells/Monocyte-derived macrophages. Seurat (v.2.3.4) and Scanpy (v.1.4) were used for data preprocessing and Uniform Manifold Approximation and Projection (UMAP). Violin plots were generated in ggplot2 (v. 3.2.1).

### 2.10. Immunohistochemistry

Four μm acetone-fixed cryosections of human or mouse liver tissue were used for histology. PBS was used to wash the sections between each step. A rabbit anti-human/mouse OPG antibody (1:200, Antibodies Online), a mouse anti-human αSMA antibody (1:500, Sigma-Aldrich), a polyclonal goat anti-collagen I (1:100, Southern Biotech, Birmingham, AL, USA), or, UK) were used, followed by secondary peroxidase-labeled goat anti-rabbit IgG or alkaline phosphatase-labeled rabbit anti-mouse IgG incubation and detection with NovaRED (Vector Laboratories, Burlingame, CA, US) or BCIP/NBT (Enzo Life Science, Farmingdale, New York, NY, US) as staining substrates. A negative control was added to each staining series by following the exact same procedures but omitting the first antibody only. OPG and collagen I expressions were quantified using Aperio ImageScope software (Leica Biosystems, Wetzlar, Germany), by quantifying areas with positive and strong positive staining over total area of the tissue. 

### 2.11. Statistics

All statistics were performed using GraphPad Prism 8 (La Jolla, San Diego, CA, USA). Normality of data was tested using a Shapiro–Wilk normality test for datasets n ≥ 7. If data were normally distributed, a paired or unpaired Student’s t-test was used to compare two paired or unpaired groups, respectively. Datasets that did not have a normal distribution were log-transformed to obtain normality and if data were still not normally distributed then nonparametric tests were used. For datasets n ≤ 7 a Mann–Whitney U or Wilcoxon test was used. When comparing multiple groups, a parametric one-way ANOVA with Holm-Sidak correction or nonparametric paired Friedman with Dunn’s correction was performed depending on the normality of the data. Correlations were assessed by calculating the Spearman correlation coefficient. *p* < 0.05 was considered significant. Data are presented as box-and-whisker plots with individual data points for unpaired data or before-after plots for paired data.

## 3. Results

### 3.1. Osteoprotegerin Expression Is Higher in Human and Murine Fibrotic Livers

We first quantified tissue levels of OPG in lysates of human livers and found that cirrhotic liver tissue contained significantly more OPG than control liver tissue (Figure 1A). Immunohistochemical staining of human liver tissue confirmed significantly more OPG expression in sections of cirrhotic livers (Figure 1B) than in sections of control livers (Figure 1C). In control livers, staining for OPG expression showed a scattered pattern throughout liver parenchyma suggesting a hepatic stellate cell distribution. In cirrhotic livers, this parenchymal OPG distribution was still present in areas relatively unaffected by fibrosis, but OPG staining was predominantly present in areas of fibrosis. A negative control for the staining is shown in Figure 1D. A double staining with αSMA, a marker of myofibroblasts, showed co-localization of OPG in αSMA-positive cells (Figure 1E) and arrows in Figure 1D indicate some of the double-positive cells. Note that the majority of the staining in the fibrotic areas appears to be extracellular pointing to the presence of OPG protein in excreted form or bound to extracellular matrix. To evaluate if this OPG is genuinely produced and excreted by human liver tissue, we incubated human precision-cut liver slices of control and cirrhotic livers for one hour to remove extracellular OPG from the slice tissue. We then replaced the medium with fresh medium, incubated the slices for 48 h, and measured the excreted OPG in culture medium. Similar to what we found for liver lysates, cirrhotic slices produced significantly more OPG than control slices (Figure 1F).

Using a mouse model of CCl_4_-induced liver fibrosis, we found similar results as in human livers. We measured higher serum levels as well as liver tissue levels of OPG after eight weeks of CCl_4_-induced liver injury as compared to healthy controls (Figure 2A,B). In control livers, staining for OPG expression was diffuse and no clear positive cells were seen as was seen for human liver tissue (Figure 2C). OPG expression in murine fibrotic liver tissue predominantly localized in areas of fibrosis, similar to OPG expression in human cirrhotic livers (Figure 2D). Quantification of the OPG staining confirmed the significantly higher OPG expression in fibrotic liver tissue as compared to control (Figure 2E).

### 3.2. Hepatic Stellate Cells Produce Copious Amounts of OPG

As our double staining suggested that hepatic stellate cells/myofibroblasts may be an important cellular source of liver OPG, we cultured primary human hepatic stellate cells (HHSteC) in a quiescent state in human serum albumin-coated wells and subsequently induced their activation and transformation to myofibroblast-like cells by culturing in uncoated wells and by culturing them in uncoated wells with additional TGFβ1 stimulation. Quiescent HHSteC produced copious amounts of OPG and this production was even higher when these cells transformed to myofibroblast-like cells as seen during fibrogenesis. The activation state was confirmed with increased expression of collagen 1A1 and alpha-smooth muscle actin (data not shown). However, co-incubating those activated cells with TGFβ1 did not further induce production of OPG (Figure 3A). 

Single cell RNA sequencing of murine liver cells showed OPG mRNA is exclusively produced by hepatic stellate cells (HSCs) and not by liver endothelial cells (LECs), Kupffer cells (KCs) or monocytes-derived macrophages (MDMs, Figure 3B). Treatment of mice with CCl_4_ for two weeks or four weeks resulted in a decline of OPG mRNA expression in hepatic stellate cells as compared to healthy control mice (Figure 3C). 

### 3.3. TGFβ1 Induces OPG mRNA and Protein Production in Murine Precision-Cut Liver Slices, Which Correlates with Other Markers of Fibrosis

To investigate liver-specific production of OPG during fibrogenesis in more detail, we stimulated mouse precision-cut liver slices with TGFβ1 to induce a fibrotic process. We found that liver tissue can express and produce OPG itself and additionally that TGFβ1 stimulation of liver slices resulted in significantly higher OPG mRNA expression, which correlated (Spearman ρ = 0.63, *p* = 0.02) with higher OPG protein excretion as compared to untreated control slices (Figure 4A–C). This higher production of OPG after TGFβ1 stimulation was accompanied by significantly more expression of the fibrosis-associated genes Col1α1, αSMA, Fn1, and TGFβ1 itself as compared to untreated control slices, but not HSP47 (Figure 4D–H). Moreover, OPG mRNA expression significantly correlated with mRNA expressions of Col1α1 and HSP47 (Figure 4I–J) and OPG protein expression significantly correlated with Fn1 and TGFβ1 mRNA expressions (Figure 4K–L). All treatments did not compromise viability of tissue slices as no significant decrease of ATP content was found in all treatment groups as compared to untreated controls after 48 h of incubation (Figure A1 of Appendix A).

### 3.4. OPG Treatment of Mouse Precision-Cut Liver Slices Results in Higher Expression of Fibrosis-Associated Markers through TGFβ1

To investigate the biological role of OPG in liver fibrosis, murine liver slices were treated with OPG itself and compared to the effects of incubation with positive control TGFβ1. Stimulation with 10 ng/mL OPG resulted in significantly higher mRNA expressions of Col1α1, HSP47, Fn1, αSMA, and most notably TGFβ1 as compared to controls, (Figure 5A–E). These effects were similar to stimulation with TGFβ1, which was included as a positive control, although TGFβ1 also significantly induced expression OPG mRNA, while this was a trend for treatment with OPG itself (Figure 5F).

We then investigated whether this profibrotic activity of OPG could be explained by its upregulation of TGFβ1 expression. We therefore inhibited TGFβ1 signaling by using galunisertib (LY2157299), a TGFβ1 receptor kinase inhibitor. We found that inhibiting TGFβ1-signaling by galunisertib in slices treated with TGFβ1 resulted in significantly lower expression of Col1α1, Fn1, and αSMA mRNA and a trend towards lower HSP47 mRNA expression (*p* = 0.07) compared to liver slices treated with only TGFβ1 alone (Figure 6A–E). A similar pattern was seen for slices treated with galunisertib and OPG as compared to slices only treated with OPG, with Col1α1, HSP47, αSMA being significantly inhibited, and no significant effects on Fn1 (Figure 6A–D). Galunisertib did not affect the expression of TGFβ1 mRNA expression after TGFβ1 or OPG treatment (Figure 6E) and treatment with galunisertib on its own did not affect expression of all genes as compared to nontreated controls.

To study whether the profibrotic activity of OPG is related to its inhibition of RANKL and TRAIL activities, slices were incubated with neutralizing antibodies against RANKL and TRAIL to mimic both known OPG scavenging activities. Again, mRNA expressions of Col1α1, HSP47, Fn1, αSMA, and TGFβ1 were used as outcome parameters. We found that incubation with the combination anti-RANKL and anti-TRAIL neutralizing antibodies more or less mimicked the results found for treatment with OPG. OPG treatment resulted in significantly more Col1α1, Fn1, and αSMA, with trends towards more HSP47 and TGFβ1 mRNA, while anti-RANKL/TRAIL treatment resulted in significantly more Col1α1, HSP47, and αSMA, with trends towards more Fn1 and TGFβ1 mRNA (Figure 7A–E). 

### 3.5. OPG Expression Responds to Spontaneous and Drug-Induced Fibrosis Resolution

To study the response of OPG production to resolution of fibrosis we used two mouse models of fibrosis resolution (spontaneous and drug-induced). In mice with CCl_4_-induced liver fibrosis, we induced spontaneous resolution by cessation of CCl_4_ for one week after four weeks of CCl_4_ treatments and we induced resolution by treatment with the antifibrotic cytokine IFNγ in the last two weeks of eight weeks of CCl_4_ treatments. Both methods of resolution induction resulted in significantly lower collagen type I deposition in liver tissue as was published by us before [28,29]. Importantly, the lower amount of collagen deposition, shown in these previous studies, was accompanied by a concomitant significantly lower OPG expression in liver tissue (Figure 8A–E). In fact, the staining for OPG, as shown in Figure 8C,D, shows that the staining pattern of OPG in liver tissue undergoing resolution of fibrosis is starting to resemble the pattern in healthy control livers again (as depicted in Figure 2C).

## 4. Discussion

OPG has previously been associated with liver fibrosis and has been included in a panel of serum markers to assess liver fibrosis severity [8,11]. We now show that it is produced in human healthy liver tissue by hepatic stellate cells and by scar tissue-associated myofibroblasts in cirrhotic livers. In mice, hepatic stellate cells produce *TNFRSF11b* mRNA, but these cells do not stain positive for OPG protein. Similar to human tissue, however, scar tissue-associated cells appear to be the main producers of OPG protein in fibrotic liver tissue. Modeling early fibrosis in precision-cut liver slices, we show that OPG expression is stimulated by exposure to TGFβ1. Moreover, OPG itself appears profibrotic through neutralizing RANKL and/or TRAIL and upregulation of TGFβ1 expression and may therefore be a novel target for pharmacological treatment. In addition, we have shown that OPG production decreases when fibrosis resolves, suggesting it may also be used as (serum) biomarker to assess treatment success.

Conflicting reports about liver-specific production and the cellular source of OPG have been published before. *TNFRSF11b* mRNA was shown in liver tissue before but it did not correlate with the higher levels of OPG found in the serum of patients with primary biliary cirrhosis compared to controls, suggesting the excess OPG is coming from a different source [11]. We therefore first assessed whether the previously reported elevated levels of OPG in liver fibrosis were a result of elevated production within the fibrotic liver or may be coming from other parts of the body [8,9]. We indeed found higher levels of OPG in human liver tissue lysates of cirrhotic livers as compared to control livers, which may point at liver production. To confirm OPG was genuinely produced in human liver tissue we subsequently used human liver slices of both control and cirrhotic livers and we found that cirrhotic slices indeed produced more OPG. This proved that liver tissue itself is capable of producing OPG under the influence of a fibrotic stimulus.

The next question was the cellular source of OPG production. An older publication by Moschen et al. has shown OPG protein expression in mainly hepatocytes through immunohistochemistry [13]. We could not reproduce this staining pattern as we only found hepatic stellate cell-associated staining in control livers and additionally scar tissue-associated staining in cirrhotic livers. Our staining pattern, however, was confirmed in a recent proteomics publication by Olander et al. showing OPG is exclusively produced in hepatic stellate cells in control livers and does not appear in hepatocytes [23]. In addition, our experiments with HHSteC indicate that hepatic stellate cells can produce OPG and activated HHSteC even more so.

Interestingly, our single cell sequencing data of mouse control livers showed that *TNFRSF11b* mRNA is exclusively produced in hepatic stellate cells, but neither we by staining nor Azimifar et al. or Ding et al. by proteomics could show OPG protein produced by hepatic stellate cells in healthy mouse livers [33,34]. In contrast, in fibrotic murine livers, our staining shows clear OPG protein expression in scar tissue-associated cells, whereas our single liver cell sequencing data show a loss of *TNFRSF11b* mRNA production by hepatic stellate cells during activation. There are several possible explanations for this puzzling finding. Firstly, it seems evident that in the case of OPG, mRNA expression does not guarantee protein expression and different regulation mechanisms appear at play here. Hepatic stellate cells may express OPG protein upon activation but may lose this ability when mRNA levels are depleted, for instance. Secondly, other cells may develop/move into the liver that produce the copious amounts of OPG we found in fibrotic liver tissue. The cells isolated for single cell sequencing were enriched for endothelial cells, macrophages, monocytes, and hepatic stellate cells and we may, therefore, have missed the cells that start producing excess amounts of OPG during fibrosis, i.e., myofibroblasts of other origins such as fibrocytes or fibroblasts. The fact that we see more OPG production by liver slices after stimulation with TGFβ1 does suggest that these cells are originating from the liver, but a contribution of, for instance, circulating fibrocytes in vivo cannot be excluded. 

Importantly, using the data generated by single liver cell sequencing, and our immunohistochemistry stainings, we found no evidence for OPG production by Kupffer cells or endothelial cells. This is in contrast with work from Sakai et al. who showed low-level production by Kupffer cells after TNFα stimulation [35]. Our single cell sequencing data did not pick up any OPG expression in Kupffer cells, but we did not specifically stimulate with TNFα, which may explain the differences found. 

An important question that arises is why hepatic OPG increases during fibrogenesis of the liver. Therefore, we first assessed whether OPG can be induced by TGFβ1, the master regulator of fibrotic responses. This was clearly the case and OPG expression was associated with other markers of fibrosis. We then assessed whether OPG itself exerts biological activities on liver tissue. We found higher mRNA expressions of all fibrosis markers tested after treatment of liver slices with OPG, although the extent of the profibrotic effect of OPG was more variable than TGFβ1. This suggests that OPG has profibrotic activities in the liver and may, therefore, be involved in generating a profibrotic environment. The next question we addressed was how OPG could exert such a profibrotic effect. Our studies with galunisertib indicate that the upregulation of TGFβ1 expression by OPG is most probably responsible for this effect as galunisertib, a TGFβ1 receptor kinase inhibitor, completely abolished the profibrotic effect of OPG. That left us with the question of how OPG can upregulate TGFβ1 expression. To date, no receptor or signaling properties of OPG itself have been reported [36]. Therefore, we hypothesized that its TGFβ1-stimulating properties are the result of either scavenging RANKL and/or TRAIL as these ligands have been reported to bind to OPG [37]. Our results using neutralizing antibodies against RANKL and TRAIL show that these have more or less the same effects as OPG, suggesting that either or both play a role in preventing fibrosis. Interestingly, RANKL was reported to play a role in hepatic cellular repair mechanisms. Sakai et al. showed high expression of the receptor for RANKL, i.e., RANK, on hepatocytes and subsequently showed that RANKL treatment can induce cell proliferation and thus reduce liver injury in a model of ischemia/reperfusion damage [35]. Preliminary studies by us treating liver slices with RANKL seem to indicate RANKL can indeed induce proliferation of cholangiocytes/hepatocyte progenitors, which will have to be further investigated.

TRAIL is a known inducer of apoptosis through its death receptors DR4 and DR5 [34]. Expression of TRAIL was shown on activated stellate cells in cirrhotic livers but not on quiescent ones [38]. Therefore, TRAIL expression may be a mechanism to induce apoptosis of myofibroblasts after tissue repair is finished and the continued presence of myofibroblasts in no longer needed. Our studies have clearly shown that activated HHSteC and (myo)fibroblasts produce OPG and therefore may be able to prevent apoptosis and maintain fibrotic conditions through this production of OPG [39].

As OPG production appears to be closely linked to the process of liver fibrogenesis, we hypothesized that OPG may also serve as a marker to assess the resolution of the disease. To verify this, we studied OPG expression in two resolution models: one model of spontaneous resolution, and one model with IFNγ as an antifibrotic drug [28]. The induction of liver fibrosis in mice in vivo using CCl_4_ has been shown to be reversible when administration of CCl_4_ is stopped. We previously found that after four weeks of CCl_4_ administration and one-week cessation of CCl_4_ administration, the deposition of collagen in liver tissue was markedly lower than animals that had received CCl_4_ for four weeks (published in [29]). Interestingly, we now show that, at the same time, OPG expression was also lower after one week of regeneration, suggesting OPG expression decreases when the liver is regenerating. A similar pattern was also found in an experiment of eight weeks of CCl_4_ administration, in which the animals were treated with IFNγ in the last two weeks of the fibrosis-inducing period. We have previously shown that treatment with IFNγ resulted in a significant reduction of fibrosis in liver tissue, indicating resolution of disease [28]. We now show that this resolution is accompanied by less OPG expression in liver tissue. These in vivo results suggest that OPG expression is not only linked to fibrosis development, but that it can also be used to detect resolution of disease. This makes it a tempting candidate for biomarker studies as the development of novel drugs against liver fibrosis is severely hampered by the lack of biomarkers to detect the efficacy of treatments [40].

## 5. Conclusions

Our data show that liver fibrosis is accompanied by higher production of OPG in liver tissue, particularly in response to TGFβ1. Hepatic stellate cells and scar-associated myofibroblasts appear to be an important source of OPG in human tissue, while in murine liver tissue scar-associated cells appear to be the main source. Furthermore, OPG has profibrotic abilities through neutralization of RANKL and/or TRAIL and upregulation of TGFβ1 expression. Spontaneous or drug-induced resolution of fibrosis is accompanied by lower expression of OPG. We therefore conclude that OPG is a liver-specific protein that is produced in response to a profibrotic stimulus and may be a novel drug target and/or biomarker for liver fibrosis. 

## Figures and Tables

**Figure 1 pharmaceutics-12-00471-f001:**
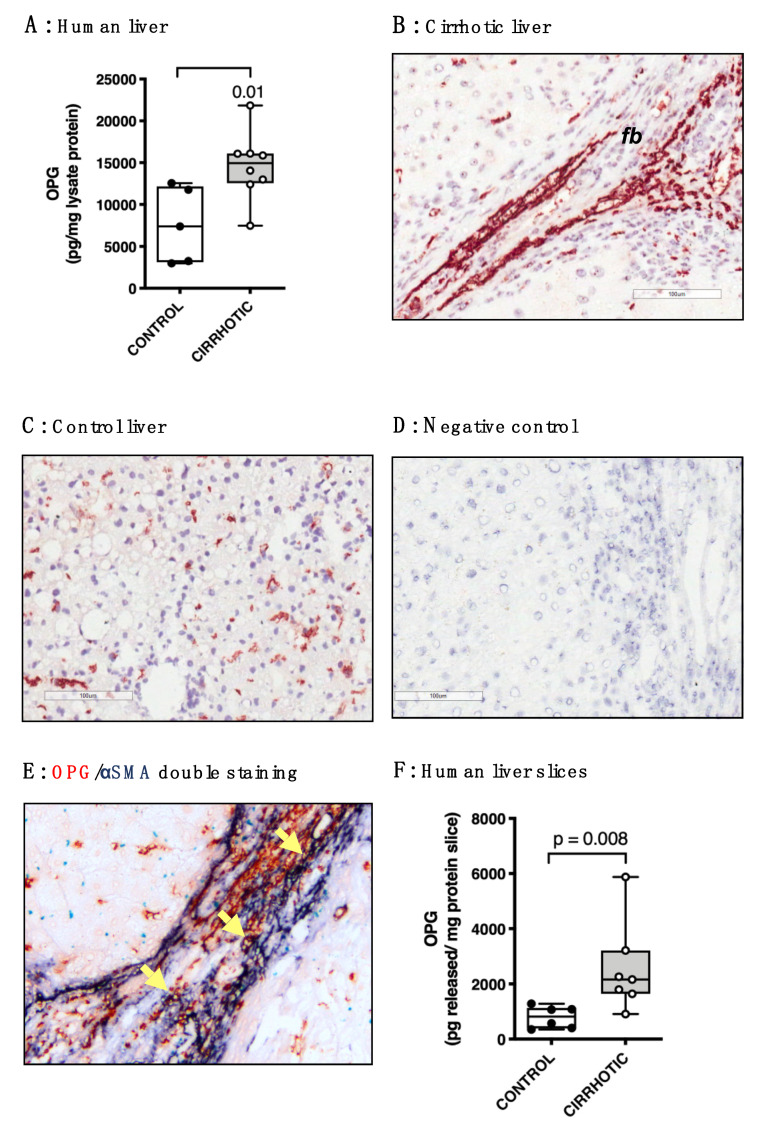
Osteoprotegerin (OPG) levels are higher in human cirrhotic livers. (**A**) Lysates of human cirrhotic livers (n = 8) contained significantly more OPG than control livers (n = 5). (**B**) Immunohistochemical staining showed pronounced OPG expression (bright red staining) in fibrotic bands (fb) and expression scattered throughout relatively unaffected parenchymal tissue in human cirrhotic livers (200× magnification). (**C**) OPG expression in control livers was only found scattered throughout the parenchyma (200× magnification). (**D**) Negative control for the OPG staining (200× magnification). (**E**) The expression of OPG (red staining) in fibrotic bands in human cirrhotic livers appears to be both extracellular and co-localizing with α-smooth muscle actin (αSMA)-positive myofibroblasts (blue staining). Some of those double-positive cells are indicated by yellow arrows (400× magnification). (**F**) Precision-cut slices of human cirrhotic liver tissue (n = 7) produced significantly more OPG in 48 h of incubation than slices of control liver tissue (n = 6). Groups were compared using Mann–Whitney U, *p* < 0.05 was considered significant.

**Figure 2 pharmaceutics-12-00471-f002:**
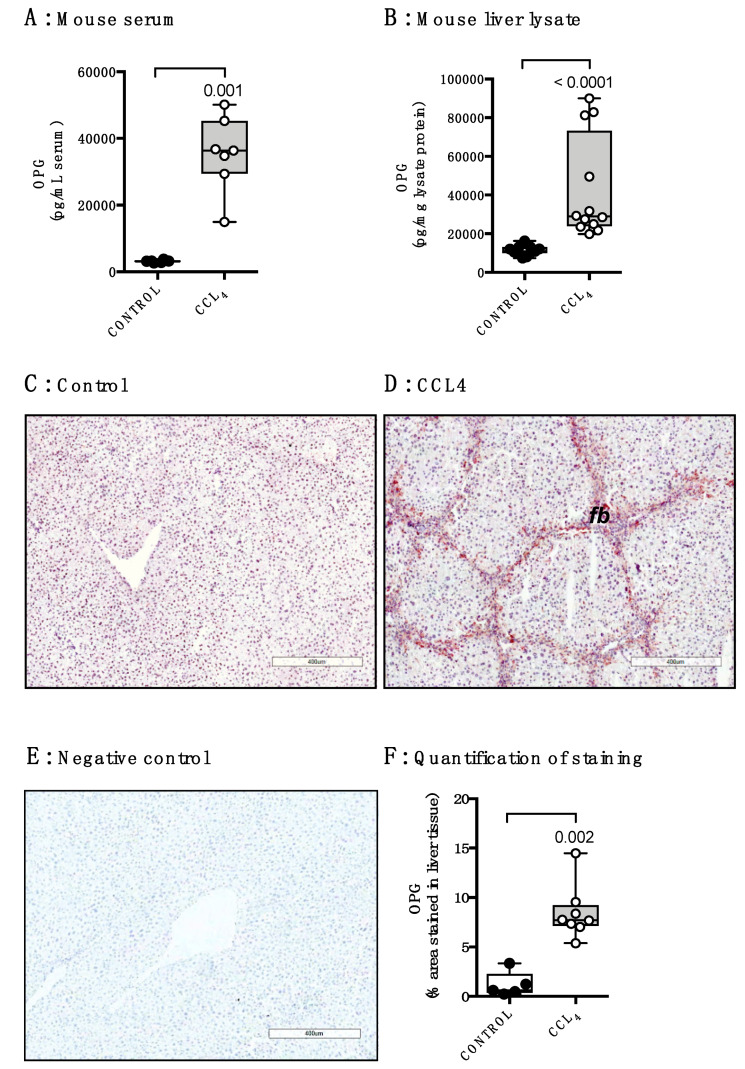
Osteoprotegerin levels are higher in murine fibrotic livers. OPG levels were higher in serum (**A**) and liver tissue lysates (**B**) of mice treated with CCl_4_ for 8 weeks compared to untreated control mice. (**C**) Immunohistochemical staining of OPG expression in control mouse livers showed diffuse staining and no clear positive cells (50× magnification). (**D**) In CCl_4_-treated mouse livers pronounced OPG expression in fibrotic bands (fb) was found (50× magnification). (**E**) Negative control for the OPG staining (50× magnification). (**F**) Quantification of this OPG staining showed higher expression in CCl_4_-treated livers as compared to control. Groups were compared using Mann–Whitney U, *p* < 0.05 was considered significant.

**Figure 3 pharmaceutics-12-00471-f003:**
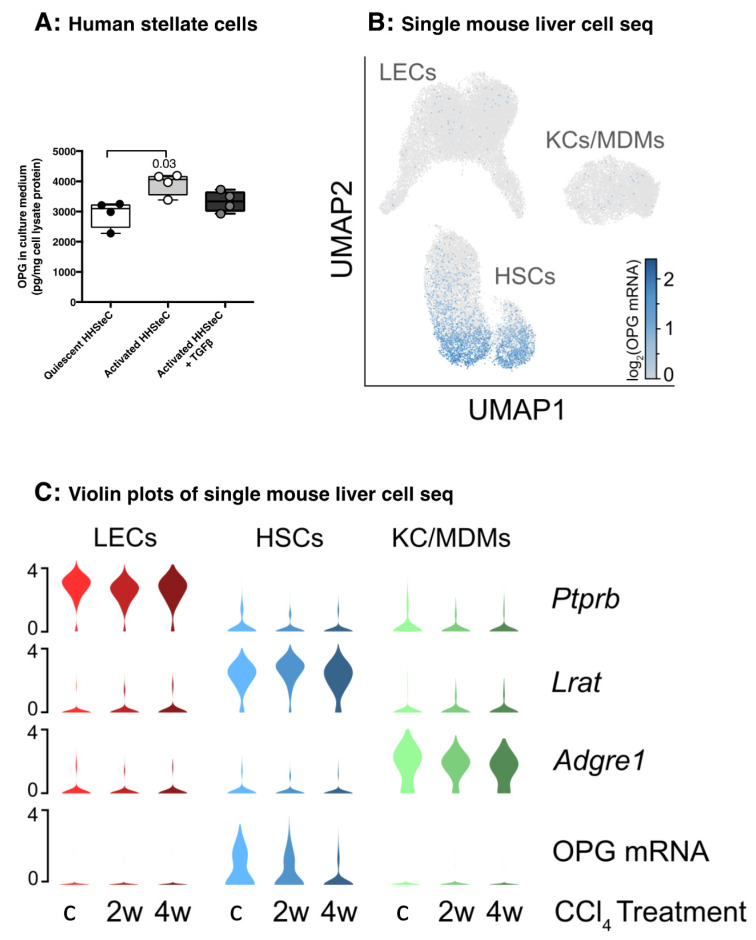
Human and murine hepatic stellate cells produce OPG. (**A**) Activated human hepatic stellate cells (HHSteC) produced significantly more OPG than quiescent HHSteC, but not when they were also treated with TGFβ1. HHSteC were activated by culturing on uncoated plastic. Groups were compared using a Friedman test with a Dunn’s correction for multiple testing, and *p* < 0.05 was considered significant. (**B**) Uniform Manifold Approximation and Projection (UMAP) showing murine hepatic stellate cells (HSCs), liver endothelial cells (LECs), and Kupffer cells (KCs)/Monocyte-derived macrophages (MDMs) overlaid with log_2_ OPG mRNA expression. The minor HSC population to the right correspond to quiescent HSCs mainly derived from vehicle-treated control mice. (**C**) Log_2_ expression of OPG and cell type markers lecithin retinol-acyltransferase (Lrat), protein tyrosine phosphatase receptor type B (Ptprb) and glycoprotein F4/80 (Adgre1). OPG mRNA expression in HSCs was less after treatment with CCl_4_ for 2 weeks (2w) or 4 weeks (4w) compared to vehicle-treated mice (c).

**Figure 4 pharmaceutics-12-00471-f004:**
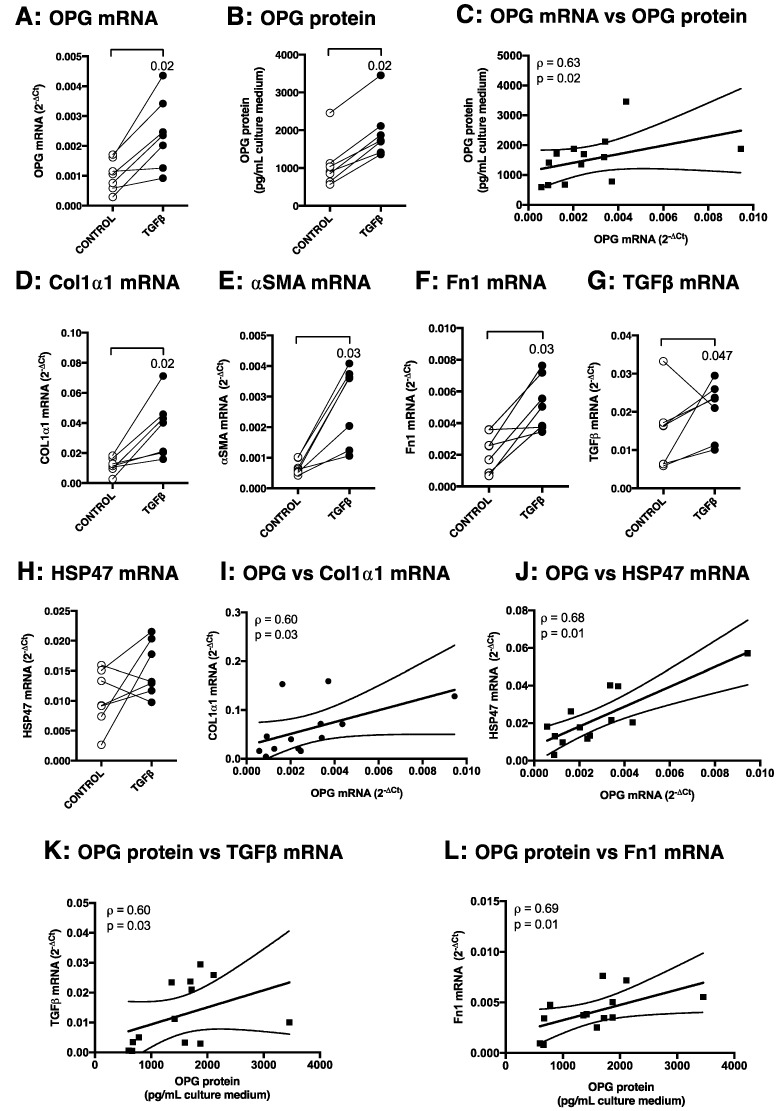
TGFβ1 induces OPG expression as well as various fibrosis-associated markers in murine liver slices. When murine liver slices were treated with 5 ng/mL TGFβ1, significantly more OPG mRNA (**A**) and OPG protein excretion (**B**) were observed, which correlated closely with each other (**C**, Spearman ρ = 0.63, *p* = 0.02). This higher OPG production was accompanied by higher mRNA expression levels of fibrosis-associated markers Col1α1 (**D**), αSMA (**E**), Fn1 (**F**), and TGFβ1 (**G**), but not HSP47 (**H**). OPG mRNA expression correlated closely with Col1α1 (**I**) and HSP47 (**J**) mRNA expression, while OPG protein secretion correlated with TGFβ1 (**K**) and Fn1 (L) mRNA expression. Groups were compared using Wilcoxon, correlations were calculated using a Spearman correlation test, and *p* < 0.05 was considered significant. For the correlations, data from control and TGFβ1-stimulated slices of Figure 5 were also added.

**Figure 5 pharmaceutics-12-00471-f005:**
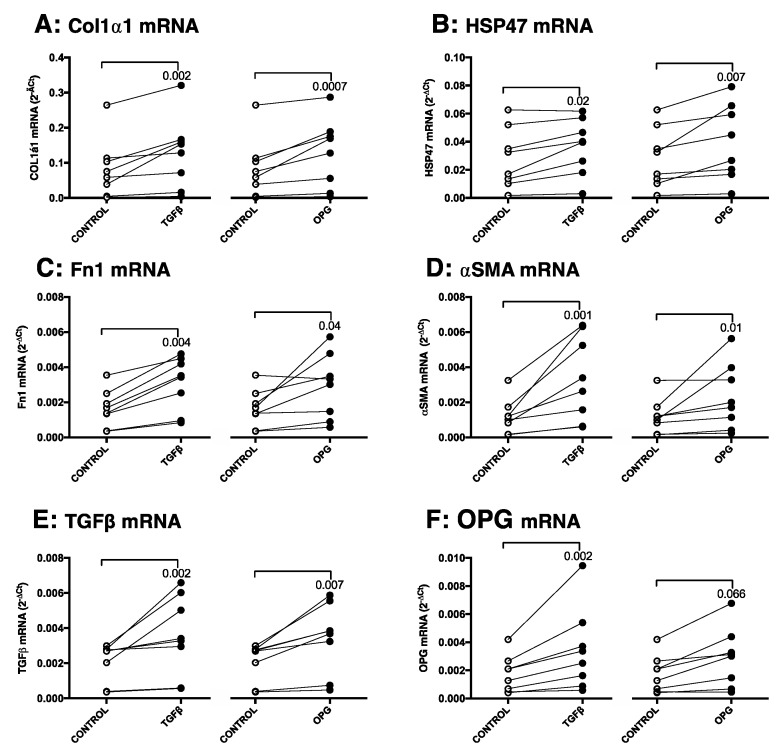
OPG treatment of liver slices results in higher expression of fibrosis-associated markers. Treatment of mouse precision-cut liver slices with 10 ng/mL OPG resulted in higher mRNA expressions of fibrosis-associated markers Col1α1 (**A**), HSP47 (**B**), Fn1 (**C**), αSMA (**D**), and TGFβ1(**E**) just like treatment with positive control TGFβ1. OPG itself was only significantly upregulated by TGFβ1 treatment and not significantly after OPG treatment (**F**). Groups were compared using a Friedman test with a Dunn’s correction for multiple testing and *p* < 0.05 was considered significant.

**Figure 6 pharmaceutics-12-00471-f006:**
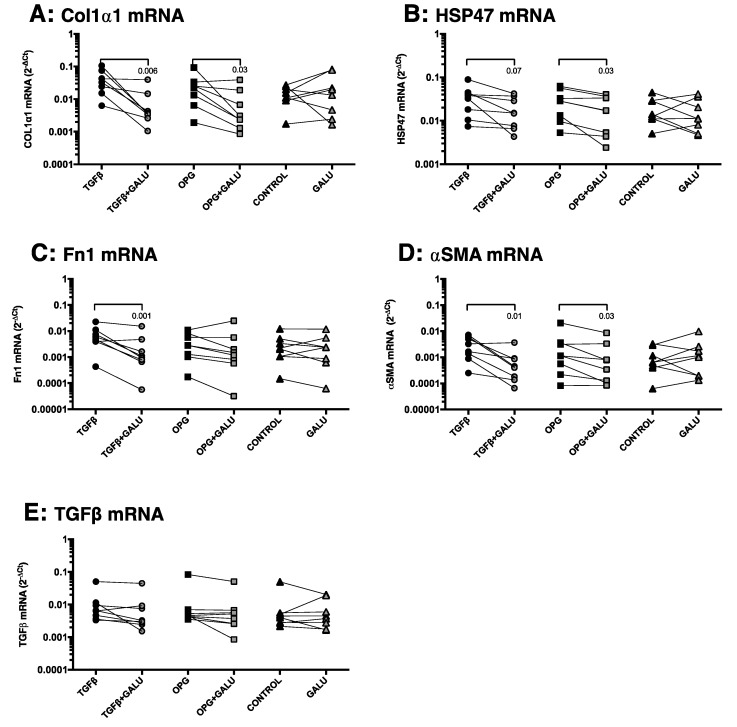
Inhibition of TGFβ1-signaling results in lower expression of fibrotic markers in slices treated with TGFβ1 or OPG. Inhibition of TGFβ signaling by treating with galunisertib (Galu) abrogated the profibrotic effects of both TGFβ1 (positive control) and OPG. Groups with parametric data were compared using a one-way ANOVA with a Holm-Sidak’s correction for multiple testing and groups with nonparametric data with a Friedman test with a Dunn’s correction for multiple testing and *p* < 0.05 was considered significant.

**Figure 7 pharmaceutics-12-00471-f007:**
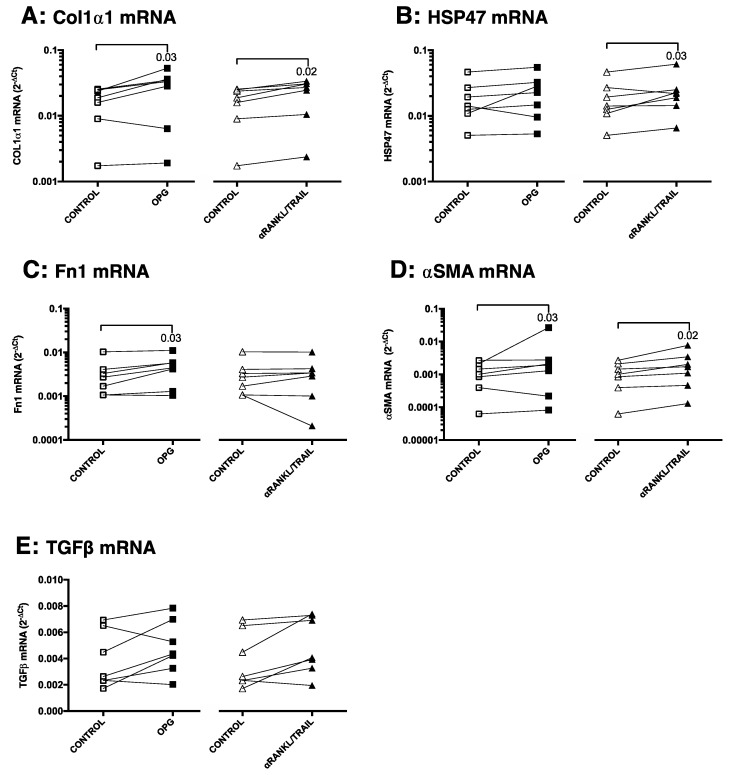
Treatment with neutralizing antibodies against RANKL and TRAIL resembles the effects of OPG. Groups were compared using a Friedman test with a Dunn’s correction for multiple testing, and *p* < 0.05 was considered significant.

**Figure 8 pharmaceutics-12-00471-f008:**
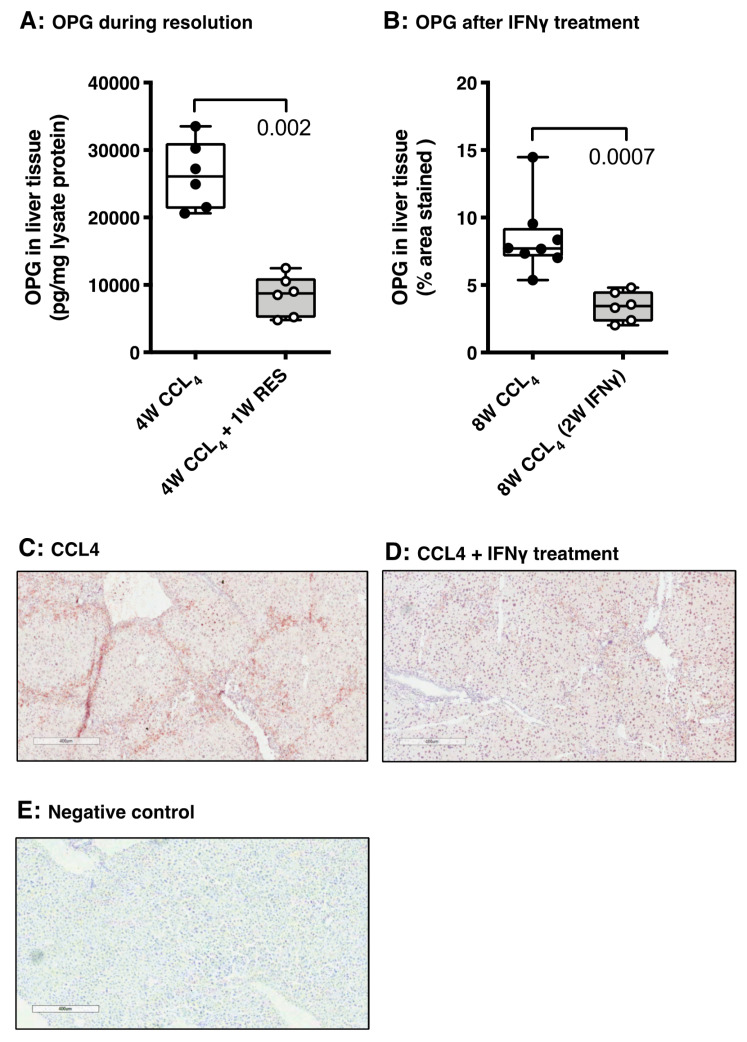
OPG responds to spontaneous and drug (IFNγ)-induced resolution. In vivo spontaneous (**A**) and interferon gamma (IFNγ)-induced (**B**) resolution of CCl_4_-induced liver fibrosis in mice resulted in lower OPG levels in liver tissue as compared to their respective controls, which was accompanied by lower collagen type I deposition as was published by us before [28,29]. (**C**) Example of the OPG staining in liver tissue of mice with CCl_4_-induced liver fibrosis that was quantified in panel B (magnification 100×). (**D**) Example of the OPG staining in liver tissue of mice with CCl_4_-induced liver fibrosis treated with IFNγ for 2 weeks that was quantified in panel B (magnification 100×). (**E**) Negative control for the OPG staining (100× magnification). Groups were compared using Mann–Whitney U, *p* < 0.05 was considered significant.

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
