# Peer review of "Osteoprotegerin Is more than a Possible Serum Marker in Liver Fibrosis: A Study into Its Function in Human and Murine Liver"

_pharmaceutics, 2020, doi:10.3390/pharmaceutics12050471_

Round 1

Reviewer 1 Report

Thank you for the authors' comprehensive revision of this manuscript. My question has been addressed in the revised manuscript.

Author Response

We thank the reviewer for accepting our revisions.

Reviewer 2 Report

I do congratulate authors for this beautiful and very comprehensive work on osteoproteregin and its relation with TGFB in liver fibrosis. I think methods and presention are excellent and fully support the results and conclusions raised. From my part I would not add anything else. 

Author Response

We thank the reviewer for accepting our revisions.

This manuscript is a resubmission of an earlier submission. The following is a list of the peer review reports and author responses from that submission.

Round 1

Reviewer 1 Report

In the present manuscript, Adhyatmika and colleagues evaluated the potential role of osteoprotegerin as a serum/hepatic marker of fibrosis, and in the pathophysiology of liver fibrosis. The topic of the study is not novel since previous studies already demonstrated its usefulness as biomarker & its biological/molecular pathways, the current study adds limited data on this.

Major concerns:

  • Novelty: The manuscript validates previous works, novelty is limited.
  • Methods: OPG stainings, please show negative staining in Figure 1 and 2.
  • Methods: OPG expression/release should be shown using the same techniques in all figures. As example, figure 8 shows OPG expression using 2 different techniques, thus does not allow a comprehensive analysis of results.
  • Methods: sc sequencing. Authors describe enrichment of cells (previous to perform rnaseq) using the F4/80 marker. This strategy leads to the selection of a sub-population of cells (rich in macrophages), thus limiting the value of the results. Additional experiments to elucidate the source of OPG during fibrosis progression are needed. These new data would increase the novelty/relevancy of the results.
  • Methods: PCLS experiments, especially those aimed at understanding the role of OPG in liver fibrosis, have to be done in fibrotic tissues (not in control).
  • Results: Figure 1. OPG staining should be quantified (as it is in Figure 2). Data about patients should be included.
  • Results: scRNAseq experiments elucidating the cell types expressing and releasing OPG should also include a possible correlation with serum OPG levels. This would help to further understand the dynamics of expression vs secretion.
  • Results: figure 8, please add representative images of OPG IHC. It would also be interesting to see the OPG expression levels in control liver tissues, to ascertain whether its expression normalizes or partly recovers. How is the expression of OPG in serum upon fibrosis regression?

Reviewer 2 Report

The authors reported that osteoprotegerin (OPG) is more than a serum marker in liver fibrosis but maybe a candidate for anti-fibrotic therapy in this manuscript. This is a well-designed study, including cell, animal and human study. However, the emerging evidences still can not convince readers about OPG had pro-fibrotic effect and own the potential of anti-fibrosis in cirrhotic liver. Two important issues should be addressed:

  1. The author concluded that OPG can up-regulated TGF-β1 to initiate or maintain fibrotic enviroment, evidencing by anti-TGF-ß1 treatment completely abolished the profibrotic effect of OPG. However, it still lacks of direct evidence that anti-OPG improves fibrosis or enhancement of OPG exacerbates liver cirrhosis.
  2. Guanabens et al. have ever reported that high OPG serum levels in primary biliary cirrhosis were associated with disease severity but not with the hepatic mRNA expression (J Bone Miner Metab 2009:27(3):347-54). They found that hepatic OPG mRNA levels were similar between cirrhotic patients and normal control. This result is contrary to the present report. The discrepant result should be discussed.